# Agronomic Biofortification with Selenium in Tomato Crops (*Solanum lycopersicon* L. Mill)

**Foroughbakhch Pournavab Rahim** [1] , **Castillo Godina Rocio** [2],
**Benavides Mendoza Adalberto** [2] , **Salas Cruz Lidia Rosaura** [3,*]
**and Ngangyo Heya Maginot** [3,*]

1 Departamento de Botánica, Facultad de Ciencias Biológicas, Universidad Autónoma de Nuevo León, Av. Universidad s/n Cd. Universitaria, San Nicolás de los Garza 66451, Nuevo León, Mexico; rahim.forough@gmail.com

2 Departamento de Horticultura, Universidad Autónoma Agraria Antonio Narro, Calzada Antonio Narro 1923, Saltillo 25315, Coahuila, Mexico; rociogcg@hotmail.com (C.G.R.); abenmen@gmail.com (B.M.A.)

3 Facultad de Agronomía, Universidad Autónoma de Nuevo León, Francisco Villa s/n, Col. Ex-Hacienda "El Canadá", Escobedo 66050, Nuevo León, Mexico

* Correspondence: biolidiasalas@yahoo.com.mx (S.C.L.R.); nheyamaginot@yahoo.fr (N.H.M.)

**Abstract:** Biofortification aims to increase the concentration of bioavailable elements in crops, to increase their nutritional quality. Selenium is a trace element of great impact on the antioxidant metabolism of plants and its accumulation is poor in species such as *Solanum lycopersicon*, so adding it is part of biofortification programs. The present work analyzes the capacity of sodium selenite ($Na_2SeO_3$) to increase the concentration of Selenium in tomatoes plants and fruits. For this, three treatments were applied (0, 2, and 5 mg $L^{-1}$ of sodium selenite) using irrigation water as a vehicle. 40 days after transplanting, the accumulation of selenium and macronutrients in leaves, stems, and fruits was quantified, as well as their impact on tomato plant productivity. Agronomic variables such as height (cm), diameter (mm) of stems, number and weight (g) of fruits produced were determined. The results were analyzed by ANOVA and later, a Tukey mean comparison test was performed. An increase in the accumulation of Se was observed, being up to 53% in the fruits under the 5 mg $L^{-1}$ treatment compared to the control. However, this increase did not have a noticeable impact on macronutrient content and tomato yield, but rather, contributed to the improvement of the nutritional quality of the tomato.

**Keywords:** selenium biofortification; sodium selenite; *Solanum lycopersicon*; health and nutrition

## 1. Introduction

Biofortification is a process of increasing the content of vitamins and minerals in a crop, through plant breeding, transgenic techniques, or agronomic practices [1]. This process is a way to improve human health and nutrition, since vitamins and minerals deficiencies are source of many diseases, that affect more than two billion individuals in the world, corresponding of one from each three people, globally [2]. Such deficiencies occur when intake and absorption of vitamins and minerals are too low to sustain good health and development [1].

During the last five decades, agricultural researches have been carryied out to increase production and availability of calorically nutrient-rich foods and improve dietary diversity, and to reduce nutrient deficiencies. According to Bouis et al. [3], consuming biofortified crops can help address nutrient deficiencies by increasing the daily adequacy of nutrient intakes among individuals throughout the lifecycle.

One of the most consumed crops around the world is tomato, that belongs to the Solanaceae family, including more than 3000 species. The *Solanum lycopersicum* L. Mill is an important source of nourishment and is part of our diet [4]. In the last few years, tomato consumption has further increased since tomato fruits supply both fresh market and processing products such as soups, juices, purees, and sauces [5]. Tomato fruits are an important source of substances with known beneficial effects on health, including vitamins, minerals, and antioxidants [6]. Indeed, tomato fruit consumption has been associated with a reduced risk of inflammatory processes, cancer, and chronic non-communicable diseases (CNCD), including cardiovascular diseases (CVD) such as coronary heart disease, hypertension, diabetes, and obesity [5]. Regular consumption of tomato fruit and its products is associated with lower risk of CNCD and several types of cancer and inflammation because of interaction of phytochemicals with metabolic pathways that are related to inflammatory response and oxidative stress [7].

However, the chemical composition of tomatoes is affected by many factors such as genetic (cultivar or variety), environmental (light, temperature, mineral nutrition, and air composition), and cultural practices (ripening stage at harvest and irrigation system) [8,9]. In order to improve the levels of human health promoting compounds in tomato fruits, the so-called "Biofortification Programs" are being used more and more frequently, both with trace elements and with macronutrients. Selenium is an essential trace element which is present in several natural kingdoms such as humans, animals, cyanobacteria [10], and some plants. This essential micronutrient contributes to the control of water status of plants [11], prevents oxidative stress, delays senescence and promotes growth [12,13], so that its adequate intake is thought to be beneficial for maintaining human health [14]. In some plants, high levels of inorganic Se can metabolize and accumulate Se in the form of organic derivatives, which is important for the plant, because it reduces the toxicity of the chalcogen, and at the same time, bioaccumulation in edible tissues allows the production of Se-enriched foods that have use as a potential nutraceutical for humans and animals [15]. Moreover, Se biofortification may elicit the production of secondary metabolites, which may benefit human health when assumed with the diet [16–18]. Some studies suggested that low intake of Se in the diet may cause a number of diseases, including heart diseases, hypothyroidism, reduced male fertility, weakened immune system, and enhanced susceptibility to infections and cancer [19,20]. On the other hand, increasing Se content in food crops offers an effective approach to reduce the Se deficiency problem in humans and animals.

Se has several oxidation states such as selenide ($Se^{2-}$), elemental selenium ($Se^0$), selenite ($Se^{4+}$), and selenate ($Se^{6+}$). The oxidized forms of selenium ($Se^{4+}$ and $Se^{6+}$) are absorbed by plants due to their high solubility, while $Se^0$ and $Se^{2-}$ are insoluble, therefore they are hardly absorbed by plants [21]. The two forms of oxidized Se (selenite and selenite) differ in terms of absorption and mobility within the plant [22]. Selenite uses phosphate transport as an assimilation pathway [22] while selenate moves through transporters and sulfate channels [23–25]. Once absorbed, selenate tends to be detected in radical tissues in inorganic form, whereas selenite appears to rapidly form organic compounds [26,27]. It is generally considered that selenium is related to antioxidant metabolism [28] and it is known that selenite induces this activity more effectively than selenate [29].

Therefore, the present paper aims to improve the nutritional quality of tomato fruits by adding sodium selenite ($Na_2SeO_3$) as an important source of Se intake, to increase the Se levels and bioavailability in tomato crops, and determine the impact of Se biofortification on micronutrient status.

## 2. Materials and Methods

### 2.1. Study Area

The experimental work was carried out in a greenhouse of the Department of Horticulture, at the Antonio Narro Autonomous Agrarian University, located in Saltillo, Coahuila, Mexico, within the coordinates 1743 m above sea level, 25°24′ north latitude and 100°02′ de west longitude of the

Greenwich Meridian. Maximum and minimum temperature and relative humidity in the greenhouse were 25 and 18 °C and 50 and 65%, respectively.

According to the Koppen climatic classification, modified by García [30], the climate is of the BSokx '(w) (e') type: Dry-arid and temperate, with a long cool summer, and a low rainfall regime throughout the year, tending to rain more in the summer and extreme weather. The average annual temperature is 22.6 °C, with an average annual rainfall of 361.4 mm, and the average annual evaporation oscillates between 1956 mm.

### 2.2. Experimental Design and Samples Preparation

Tomato seeds from the *Rio Grande* variety released by the EDENA Seed Company in the USA, were seeded in 200 cavities polyethylene trays, under a black polypropylene mesh, to provide shade. 40 days later, the seedlings most similar in size and development were selected for transplantation in 20 L pots, with the application of a rooting solution.

The substrate used was a mixture of peat-moss and perlite in the ratio 70:30, and three treatments with anhydrous sodium selenite ($Na_2SeO_3$) of the respective concentrations 0, 2 and 5 mg $L^{-1}$, as sources of selenium (Se) were applied. Crop nutrition was performed through irrigation at field water capacity every three days, with a Steiner's universal nutrient solution adjusted to an acidic pH (between 5.5 and 6.5) with sulfuric acid, to ensure the availability of the mineral elements and maintain the selenite ion in its protonated form [31]. The lateral buds were pruned every 8 days, with a preventive application of phytosanitary products without selenium, to control pests and diseases.

### 2.3. Agronomic Yield

At 45 days after transplantation, data were recorded on the different plant components, leaves, stems and fruits, for the morphometric analysis of the studied plant. In each treatment, 15 plants were randomly selected and labeled from the start of the transplant, being 45 plants in total. The number and weight of mature fruits per plant were recorded, as well as the polar diameter and the cross diameter of these fruits, using a digital vernier, Autotec, Caliper digital 150 mm model. Also, measurements of plant height (H) were made from the base of the stem to the last leaf in the aerial part of the plant, using a tape measure, reporting the values in centimeters (cm), as well as the diameter of the stem (D) at the base of the plant, using the digital Vernier.

For the determination of dry matter, 6 plants of each treatment were randomly selected and from them, the two leaves with physiological maturity and their corresponding stems were cut, and the samples were weighed to obtain the fresh weight. They were placed in previously-labeled esterase paper bags, and dried in a drying oven at a temperature of 60 °C for 48 h. After the drying time, they were weighed again, and the value of the weight of the bags was subtracted from the fresh weight, to obtain the weight of dry matter corresponding to each component.

### 2.4. Selenium Content and Nutritional Status

Samples from the different components of the mature plants were obtained and macerated; 1 g of the macerate was taken, and the sample was digested with nitric and perchloric acid in a 3:1 ratio, using a heating plate at 100 °C. Subsequently, the solution was filtered through Whatman No. 42 filter paper and made up to a 100 mL working solution with deionized water. It was subjected to a Plasma Induction Spectrometer (ICP), brand THERMO JARELL ASH, Model IRIS Advantage, following procedure 984.27 of the AOAC [32], to obtain the value of the elements K, Mg, and Ca, as well as the selenium in each studied component.

As for nitrogen, it was quantified by the Kjeldahl method [33] and phosphorus, by the colorimetric method of the ANSA aminonaphthol sulfonic acid reagent [34]. The previously prepared digestions were used for the minerals quantification (K, Ca, Mg, Fe, Zn).

### 2.5. Statistical Data Analysis

For the agronomic variables (yield, height and diameters), as well as the quality variables (content of selenium and macroelements), an analysis of variance was performed with the statistical package SAS 9.1.3, to verify the significant differences of the variables in each treatment, at $p \leq 0.05$. A Tukey mean comparison test was applied to identify the groups formed in the different treatments.

## 3. Results

### 3.1. Yield of Agronomic Variables

The agronomic variables presented significant differences ($p \leq 0.05$) between the Se treatments compared to the control (Table 1). The height of the stem presented 2 statistically different groups, one formed by the control T0 (0 mg L$^{-1}$), with the value of 61.1 cm, and the other formed by the treatments T1 (2 mg L$^{-1}$) and T2 (5 mg L$^{-1}$), with values of 67.5 and 65.5 cm, respectively, highlighting a significant increase in height with the application of sodium selenite. Similarly, the diameter of the stem registered 2 statistically different groups, one with the control (11.8 mm), and the other constituted by the 2 treatments T1 and T2, with the values of 13.4 and 13.3 mm, respectively.

**Table 1.** Agronomic values from tomato under different sodium selenite treatments.

| Treatments | Stems | | Fruits | | Dry Matter | |
|---|---|---|---|---|---|---|
| | H (cm) | D (mm) | N (Fr/plant) | W (g) | L (g) | S (g) |
| 0 mg L$^{-1}$ | 61.1 ± 3.9 b | 11.8 ± 0.9 b | 12.00 ± 2.9 a | 869.20 ± 224.6 a | 43.7 ± 14.5 b | 26.6 ± 3.0 a |
| 2 mg L$^{-1}$ | 67.5 ± 3.0 a | 13.4 ± 0.8 a | 17.67 ± 3.9 a | 1061.53 ± 349.8 a | 48.1 ± 19.1 ab | 28.3 ± 3.6 a |
| 5 mg L$^{-1}$ | 65.2 ± 2.9 a | 13.3 ± 0.8 a | 16.67 ± 3.6 a | 1184.83 ± 378.9 a | 50.9 ± 19.2 a | 29.9 ± 6.5 a |

H: Height, D: Diameter, N: Number, Fr: Fruits, W: Weight, L: Leaves, S: Stems. In each column, means followed by different letter are statistically different ($p < 0.05$), and the standard deviation (SD) is reported after +/−.

Regarding fruits, no statistically significant differences ($p \geq 0.05$) were detected among the control and the treatments, both for number and weight. Values of 12, 18 and 17 fruits per plant were recorded for T0, T1 and T2, respectively, forming a single statistical group, although the content of Se increased (Table 1). Similarly, the weight of the fruits showed a single statistical group with the application of sodium selenite, registering the values 869, 1062 and 1185 g per plant for T0, T1, and T2, respectively (Table 1).

In the case of dry matter, 3 statistically different groups were recorded in the leaves ($p \leq 0.05$), with increasing values according to the concentration of sodium selenite applied: 43.7 for the control, 48.1 and 50.9 m for T1 and T2, respectively. The stems presented a single group, with slightly increasing values of 26.6, 28.3, and 29.9 g for T0, T1, and T2, respectively.

### 3.2. Selenium and Macronutrient Content in Different Tomato Components, under Sodium Selenite Treatments

The concentration of selenium in the different analyzed components indicated significant differences ($p \leq 0.05$) among the treatments, registering the highest distribution in the stems, with the values of 21.7, 45.6 and 52.3 µg g$^{-1}$ for T0 (0 mg L$^{-1}$), T1 (2 mg L$^{-1}$) and T2 (5 mg L$^{-1}$), respectively (Figure 1). The leaves presented the lowest Se distribution, with values of 9.9, 20.9, and 20.4 µg g$^{-1}$ for T0, T1 and T2, respectively (Figure 1). The fruits registered values of 16.8, 24.5 and 35.8 µg g$^{-1}$ for T0, T1 and T2, respectively. These results show an increase in the content of selenium with the application of sodium selenite in all the components evaluated (Figure 1).

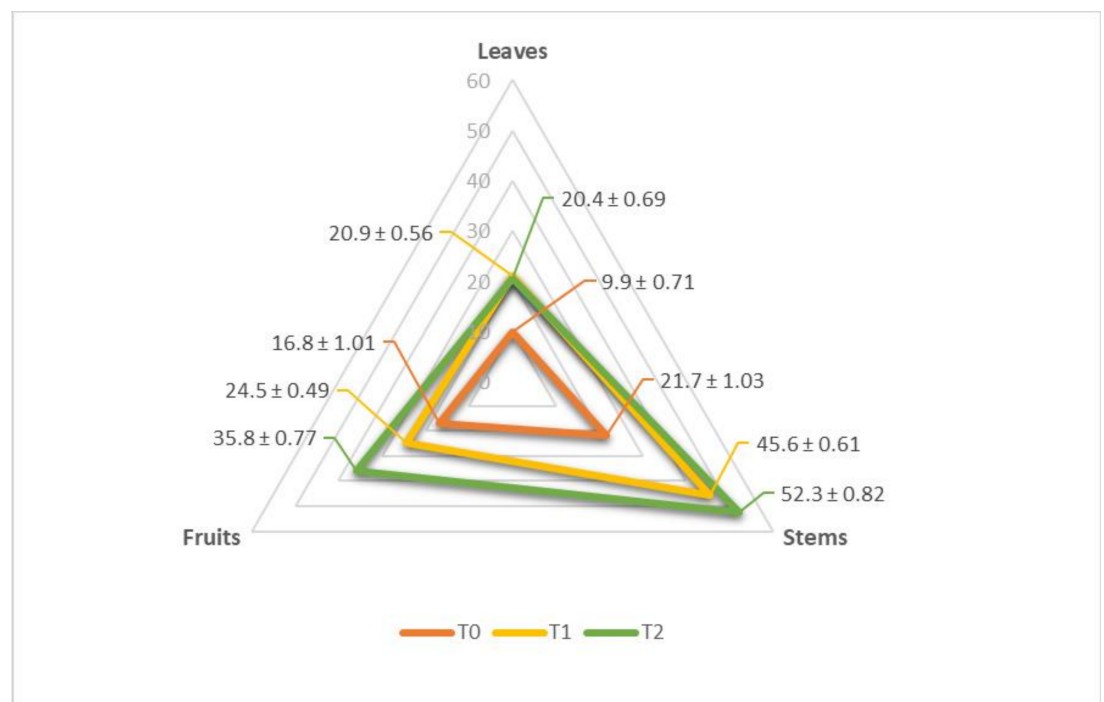

**Figure 1.** Accumulated selenium content ($\mu g\ g^{-1}$) in tomato components under sodium selenite treatments T0 = 0 mg L$^{-1}$, T1 = 2 mg L$^{-1}$, T2 = 5 mg L$^{-1}$.

The macronutrient content in the different tomato components is shown in Figure 2, presenting potassium and calcium as the elements that were statistically significant ($p \le 0.05$) among treatments, which suggests that the sodium selenite applied in this study did not interfered with the absorption of other elements. The values obtained were 3.57, 3.84, and 2.95 $\mu g\ g^{-1}$ in the control (T0 = 0 mg L$^{-1}$) for potassium in leaves, stems and fruits, respectively. In treatment 1 (T1 = 2 mg L$^{-1}$), these values were 4.42, 4.47, and 3.44 $\mu g\ g^{-1}$ in leaves, stems and fruits, respectively. For its part, calcium registered the values 3.3 and 3.9 $\mu g\ g^{-1}$ in stems for T0 (0 mg L$^{-1}$) and T1 (2 mg L$^{-1}$), respectively; 0.86, 1.26, and 2.82 $\mu g\ g^{-1}$ in fruits for T0 (0 mg L$^{-1}$), T1 (2 mg L$^{-1}$) and T2 (5 mg L$^{-1}$), respectively.

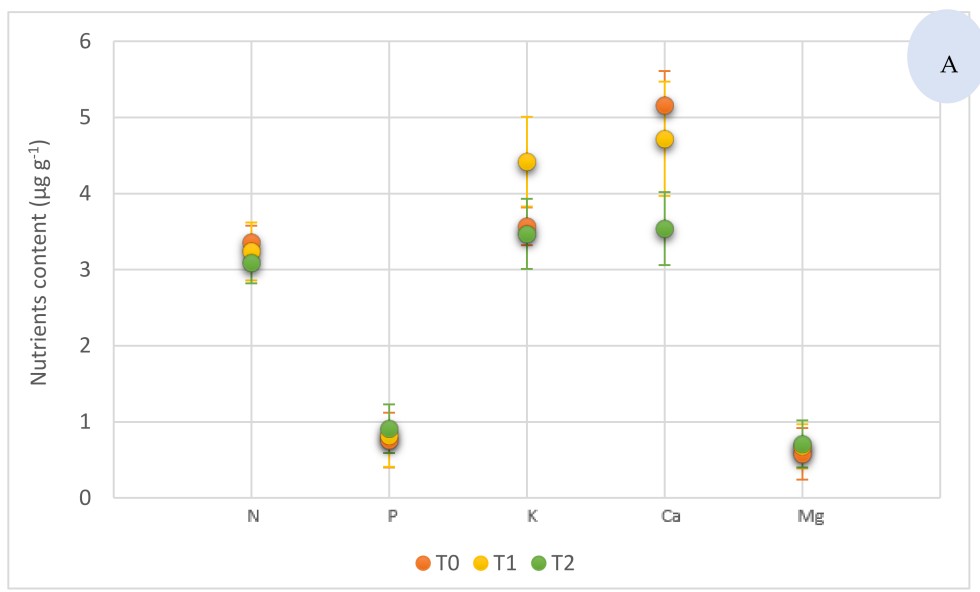

**Figure 2.** *Cont.*

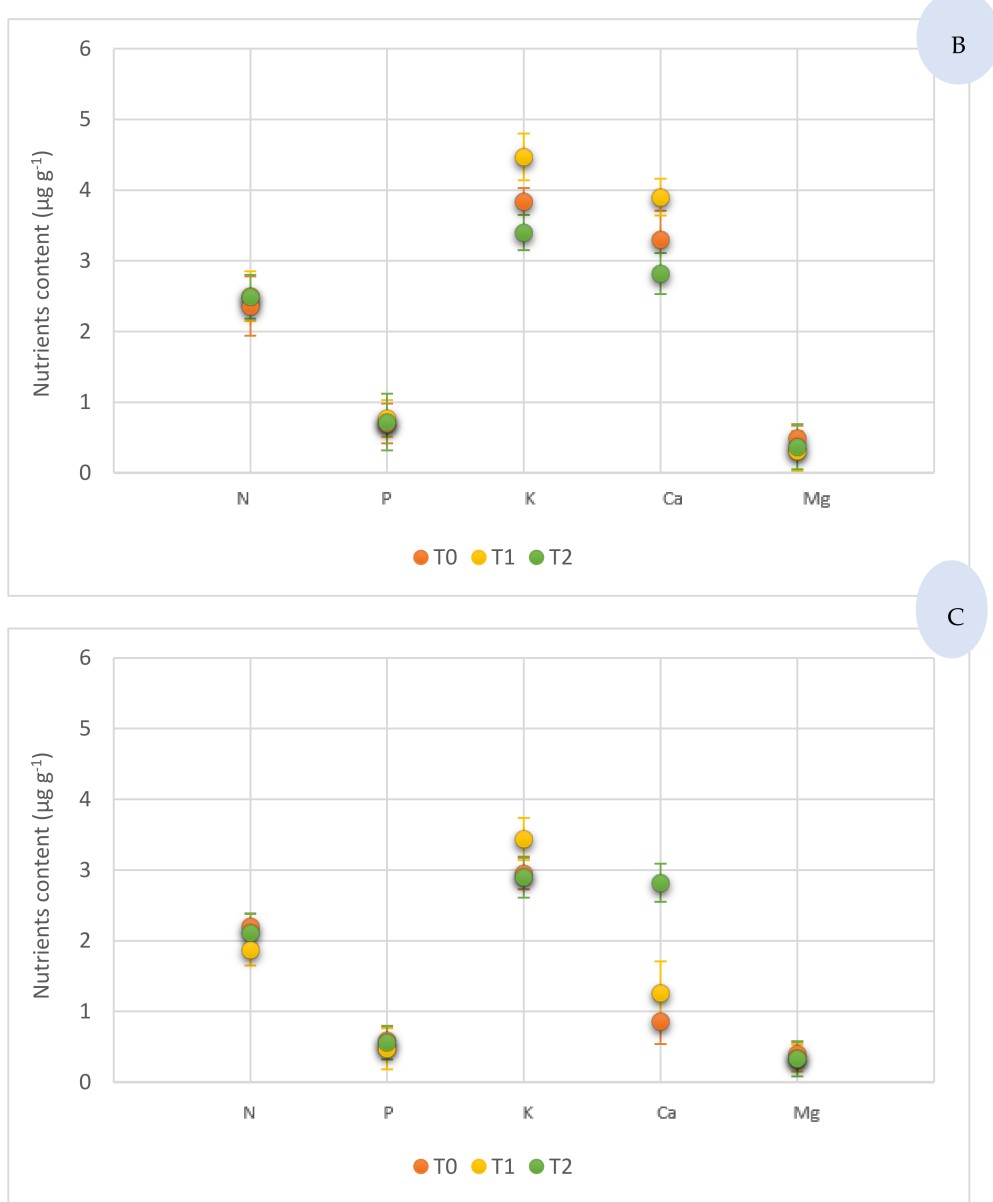

**Figure 2.** Macronutrient content in different tomato components under sodium selenite treatments. (**A**) Leaves, (**B**) Stems, (**C**) Fruits. T0 = 0 mg L$^{-1}$, T1 = 2 mg L$^{-1}$, T2 = 5 mg L$^{-1}$.

## 4. Discussion

### 4.1. Agronomic Variables of Tomato under Sodium Selenite Treatments

The agronomic variables yield, stem height (H), stem diameter (D), and fruit weight (W) were significantly increased under the application of sodium selenite (2 and 5 mg L$^{-1}$), demonstrating that Se biofortification can improve the tomato productivity. This result agrees with those of Hasanuzzaman et al. [35], who found that Se can improve plant growth and development, since it has vital roles in reducing negative consequences of abiotic stresses. This beneficial role of Se were studied by some authors, who reported that it improves plant tolerance to many abiotic stresses, as drought [36,37], salinity [38,39], cold [40,41], metals/metalloids [42,43], and UV-induced stress [44,45]. Therefore, it must be applied at relatively low concentrations, in accordance with Chauhan et al. [42], who found that low dose of Se can stimulate plant growth, improve photosynthesis and help in homeostasis of essential nutrient elements. Also, it has been shown that Selenium serves as an anti-senescent and helps in

maintenance of cellular structure and function, thus, contributes towards improved plant growth and development [46,47]. Nevertheless, Se at excessive concentrations leads to toxicity in plants, resulting in chlorosis and necrosis as well as restricted growth and reduced protein biosynthesis [48,49], so that it is essential to keep the concentration low during the Se application, to achieve the desired effect.

On the other hand, selenite remains in organic form when absorbed [26,27], and it has been shown to be a more efficient inducer of glutathione peroxidase [29], which is determining for the production of healthy tomatoes high in antioxidant nutrients. On the strength of the above, the selenium specific effect on tomato production seems to depend on the chemical form in which Se is applied, as also indicated by Germ et al. [50] and Becvort-Azcurra et al. [51], who found no differences in fruit yield when applying sodium selenite.

*4.2. Selenium Content in Different Components of Tomato under Sodium Selenite Treatments*

The selenium content registered a significant increase for leaves and stems in the applied sodium selenite treatments. In the fruits, only one notable increase was reported under the 5 mg $L^{-1}$ treatment, corresponding to 53% more selenium, compared to the control. Nancy et al. [52] reported a similar value (52.5% compared to the control), registering 29.5 $\mu g\ g^{-1}$ in tomato fruits obtained with 10 mg $L^{-1}$ of sodium selenate, applied to the soil. Other authors have reported the accumulation of Se in wheat grains [53] and rice grains [54] through Se fertigation and Se foliar application, respectively. In broccoli (*Brassica oleracea* L.), Se fertilization was shown to reduce the amount of total phenolic acids, without altering the profile distribution of specific compounds [55].

According to Kabata-Pendias and Pendias [56] who reported uneven accumulation among different organs, actively growing tissues usually contain higher amounts of Se, and many plant species accumulate higher amounts of selenium in stems and leaves than in the root tissues. The same authors noted that higher levels of selenium in plants can suppress the concentration of N in tissues and can inhibit the absorption of some metals such as Mg [56]. However, Selenium delivery in a food system depends mainly on the levels of plant available Se in soils used for agriculture [57]. The element's availability in soils depends on soil pH, redox potential, cation exchange capacity, and levels of Fe, sulfur, aluminum, and carbon [58,59].

Malagoli et al. [25] found variations in tomato S uptake and assimilation induced by Se, which caused changes in the synthesis of S-secondary compounds with nutritional value, such as glucosinolates (GLS), which function in plant defense against insects and herbivores. Because S nutrition is strictly associated with N metabolism, Se can exert an additional effect on the synthesis of proteins and amino acids, as well as on N-secondary compounds with free radical scavenging activities, like phenolics.

Much less common than Se deficiency, Se toxicity can occur as selenosis, characterized by hair loss and thickened nails, as it was the case in Enshi in the Chinese province of Hubei, by 1961 to 1964, caused by eating crops grown on high-Se soil [60]. So that, the daily recommended intake of Se is mostly 40–75 mg/day globally, with <30 mg/day inadequate and >900 mg/day potentially harmful. However, tolerable upper limits have been set lower, in the range of 400–450 mg/day for the United Kingdom, United States, Canada, EU, Australia, and New Zealand [61].

*4.3. Macronutrients Content from Different Components of Tomato under Sodium Selenite Treatments*

The results showed that the sodium selenite applied in this study did not interfered with the absorption of other elements, which suggests that there is no modification when adding selenium in the irrigation solution, since in this way, there is no antagonism of selenium with the other nutrient elements. This could be an advantage for elements such as K, which is one of the most abundant elements in plant tissues, comprising around 10% of dry matter [62]. Potassium is involved in numerous biochemical and physiological processes vital for growth, yield, quality, and stress [63]. It stands out as the cation that has the greatest influence on the quality parameters that determine the commercialization of fruits, consumer preferences, and on the concentration of associated phytonutrients of vital importance for

human health [64]. Other authors such as Smoleń et al. [65] reported a reduction in the levels of Ca and Mg in lettuce roots with the foliar application of Se and I, although they did not observe any difference in the content of macronutrients in leaves when they applied Se individually.

## 5. Conclusions

The sodium selenite added to the nutrient solution biofortified the tomato through a significant increase in selenium content for the different components evaluated (fruits, leaves and stems), producing up to twice its concentration in fruits for the 5 mg L$^{-1}$ treatment, compared to the control. The application of sodium selenite did not interfere with the absorption of macronutrients, but rather, contributed to the improvement of the nutritional quality of the tomato. Se applied in low concentration can improve crop yield and food quality.

**Author Contributions:** Conceptualization, F.P.R. and C.G.R.; methodology, C.G.R., F.P.R., A.B.M., and N.H.M.; software, C.G.R., M.N.H., and S.C.L.R.; validation, F.P.R. and B.M.A.; formal analysis, C.G.R. and N.H.M.; investigation, C.G.R., B.M.A., and F.P.R.; resources, F.P.R. and B.M.A.; data curation, N.H.M., C.G.R., and F.P.R.; writing—original draft preparation, C.G.R. and N.H.M.; writing—review and editing, F.P.R., N.H.M., B.M.A. and S.C.L.R.; visualization, F.P.R., B.M.A., N.H.M. and S.C.L.R.; supervision, F.P.R. and B.M.A.; project administration, C.G.R. and F.P.R.; funding acquisition, C.G.R. All authors have read and agreed to the published version of the manuscript.

**Funding:** This research received no external funding.

**Conflicts of Interest:** The authors declare no conflict of interest.

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
