# Peer review of "Agronomic Biofortification with Selenium in Tomato Crops (Solanum lycopersicon L. Mill)"

_agriculture, doi:10.3390/agriculture10100486_

Round 1
Reviewer 1 Report
The manuscript is interesting however has to be improved in order to add scientific value since at the present form is a technical article. Below specific comments
Abstract
l.16 replace “tomatoes” with the scientific name of the plant
- 18 add fruits as follow “of Selenium in tomatoes plants and fruits”
l.21 add plant as follow “on tomato plant productivity”
At the end of the abstract add the implication that your findings can have
- 27 Keywords replace tomato with the scientific name of the plant
Introduction
Specify the novelty of your work and highlight the difference from paper on similar topic.
- 30 replace density with content
- 42 add L. Mill
- 53-65 I suggest you to better explain the beneficial effects of Se on plants and human since it is the focus of your work. At this scope you can read and possibly cite this reviews https://doi.org/10.1021/acs.jafc.0c00172 https://doi.org/10.1007/s11130-019-00769-z
Materials and methods
Study area: add some details about climatic conditions
Experimental design: explain why you choose selenite and soil treatment and not other Se forms or treatment methods (see also comments for discussion section)
- 79 replace in with and
- 81 add were applied as follow “of selenium (Se) were applied.
Give more details about treatment methods i.e. how much water for each pot? How many times per week you give water? On the basis of what you decide the irrigation volume?
Agronomic yield: why you did not report data about roots development? I think it can be interesting since some studies find that Se enhance drought resistance in plants and this can also be related to a greter capacity of water uptake (see for example https://doi.org/10.3389/fpls.2018.01191 https://doi.org/10.1007/s12011-009-8328-7)
Selenium content and nutritional status
Specify how the samples for analysis were prepared (from how many plants etc)
Specify that is total Se content. It can be interesting also to know in which form the Se is present in the different parts of the plants (for the discussion of this parameter you can see https://doi.org/10.1016/j.jtemb.2016.11.015)
Experimental Design …
The experimental design is not very clear since at l. 85 you wrote 15 plants for each treatment and so it can be assumed that you have 15 replications for each treatment and so why at l. 103 you assess 3 replications?
However, since at l. 85 you wrote 15 plants were chosen you have to specify the total number of plants for each treatment. In addition, I suggest you to replace “chosen” since for a good statistical design you have to take the plants for the measurements randomly.
Specify the p level
Results
I suggest you to do not repeat in the text the data already present in the table/figures but to make some comments and correlation.
For the data presented in Fig. 1 and Fig. 2 you have to present also the standard error
Tab. 1 why in some cases you put *? In general, if there are different letters it can be assumed that the difference is statistical significative. Specify if after +/- you report SE on SD
Y specify that is the number of fruit per plant
Below the table you can add a sentence like In each column means followed by different letter are statistically different (p…)
Why Se increase H and diameter of plants but not the DW?
In the text add a reference to the tables and figures you are writing about
- 120-124 you can not assess that there is a gradual increase since the differences are not significative (Tab. 1)
- 132-149 why you do not evaluate the Se content of roots?
Discussion
I suggest to the authors to totally rewrite the discussion sections since in the present form they are not acceptable. In particular in this sections the authors have to compare their data with similar or in contrast findings in bibliography. Moreover in this sections they have to give a scientific explanation about the data observed. Below some detailed comments
l.160-170 so why you choose selenite and soil application?
- 184-188 I suggest to remove this parts since is not related to your work.
Divide the discussion part for Se content from the part for macronutrients
Why Se promotes plants growth?
Conclusions
Try to give some possible applications of your finding (i.e. how many tomatoes can I eat without reaching toxicity levels for Se? See for example https://doi.org/10.1002/jsfa.9030)
Highlight what your findings add to the knowledge already available on this topic
Author Response
Dear reviewer, thank you very much for your wise comments, which helped a lot to improve our manuscript, and especially for the documents that you recommended to us, with precise information that was used in this work.
Reviewer 2 Report
Comments and Suggestions for Authors
This article reports the effects of treatments of sodium selenite on Se bio fortification of tomato crops. While not highly novel, it contributes to to increasing knowledge about the bio-fortification of tomato crops. However there are deficiencies in trial description and presentation of results that make it difficult to judge the importance and real value of the study and therefore should be substantially revised. The paper would benefit from moderate editing for grammar and spelling.
Major comments:
INTRODUCTION:
The introductory literature used does not correspond to the results and purpose of the research. It is overloaded with information about human health and there is not much information about the subject of research. The authors should add the details on current scientific knowledge of tomato biofortication, highlighting the innovative contribution of this paper on this topic.
MATERIALS AND METHODS:
The materials and methods lack information on the implementation of the experiment which precludes the reproducibility of the work.
Lines 71 - 74. Better describe the structural characteristics of the greenhouse and provide information on the climatic parameters (eg temperature and relative humidity trends) in the greenhouse during the growth period of the tomato plants. Indicates the time period in which the experiment was carried out.
Lines 76 - 78. Indicates the name and main characteristics (eg type of plant growth, characteristics and use of fruits) of the tomato variety used in the experiment. Mention the trade/brand name of rooting product used.
Lines 79 - 83. Better describe selenium treatments, specifying the number and frequency. Define the amount of selenium given to each plant during the experiment. Estimate the total amount of selenium absorbed by each plant. Better describe the fertilizer solution management, indicating the quantity given to the each plant and the frequency of kind of ferirrigation. Quantify the amount of mineral elements given to the each plant during the experiment. Explain whether the leaching solution has been quantified and analyzed.
Lines 85 - 92. Indicates the phenological stage of the plants at 45 days after transplantation (e.g. full ripening or end of ripening). Explain better the meaning of “number and weight of mature fruits per plant”. It is not clear whether it is the ripe fruit at that time or all the fruit that the plant has ripened. The following section of the results (paragraph 3.1) comments on the dry matter data of leaves and stems. They cannot be found in the materials and methods section. Indicates how the dry matter of the various organs of the plant was determined.
RESULTS:
The results should be improved using appropriate agronomic terms and possibly integrated with other data that allow to better understand the effects of the different treatments. It would also be advisable to reconsider the way the results are presented (table and figures).
Lines 119 - 124. In the agronomic field, the term “yield” (or “harvest yield”) is conventionally used to indicate the quantity of fruits (eg kilograms) produced by a plant and not the number of fruits. Use the appropriate agronomic terms in the text and Table 1 for this and other parameters considered.
Lines 117 - 118. Table 1 should be supplemented with data relating to the average weight of the fruit and the dry matter content of the fruit. If there is not enough space in the table to enter all the parameters, it is advisable to make two tables, one with the vegetative parameters and one with the reproductive ones. Table 1 also lacks the indications for interpreting the statistical analysis. Enter this information.
DISCUSSION:
In paragraph 4.1, where the effects of sodium selenite applications on the agronomic parameters are discussed, the effectiveness of the different forms of selenium (sodium selenite compared to sodium selenate) is essentially analyzed. The different vegetative-productive responses found in this experiment are not explained. Explain, with the support of the literature, the reasons that determine the increase in growth and productivity of tomato plants treated with selenium, eliminating the unnecessary parts of text.
The same reasoning is valid for paragraph 4.2 where the selenium and macronutrients content in different components should be discussed in relation to the sodium selenite applications. These aspects are only partially discussed. Above all, the reasons for the different accumulation of selenium in the different organs of the plant are not fully explained. Instead, a series of results obtained in other researches is reported which often has difficult connection with the results obtained. Discuss better the results obtained, using only the strictly necessary literature, i.e. the one that helps to explain the results obtained. Delete the other parts.
Lines 157-159. Please consider revising this piece of text in a grammatically correct manner.
Lines 163 and 171. Five references in a row, seems not very appropriate. Please select any suitable references and delete the rest.
Author Response
Dear reviewer, thank you very much for your wise comments, which helped a lot to improve our manuscript.
Reviewer 3 Report
The authors investigated "the Agronomic biofortification with Selenium in tomato". The manuscript is well-written and easily readable. The topic is interesting, but not very innovative.
The authors should check the text for English form and fix the formulas. In recent years many scientific works "on biofortification" have been published, so it is advisable to include more recent references. For example:
De Bruno A., Piscopo A., Cordopatri F., Poiana M. and Mafrica R. Effect of Agronomical and Technological Treatments to Obtain Selenium-Fortified Table Olives. Agriculture 2020, 10, 284; doi:10.3390/agriculture10070284
Author Response
Dear reviewer, thank you for your wise comments and especially for the document that you recommended to us, for the improvement of our manuscript.
Round 2
Reviewer 1 Report
The manuscript was improved a lot but I still think that it needs improvement especially in the results and discussion section. The authors have to correlate the results obtained and try to give some explanation about the data observed. In the discussion section, a wider comparison with the results obtained by other authors and on the possible mechanism of action has to be reported. For your guidance see
https://doi.org/10.1016/j.envexpbot.2020.104170
https://doi.org/10.1080/10643389.2019.1598240
https://doi.org/10.1021/acs.jafc.0c00172
Author Response
The Reviewer is absolutely right.
We thank him/her for the suggested documents, which allow us to remove informations that did not fit very well with the work, to replace them with the most appropriate one.
Kind regards!
Reviewer 2 Report
The article reports results of treatments with sodium selenite on the bio fortification of tomatoes. The topic is interesting, but not very innovative. The manuscript is overall well written and quite readable. It lacks some information on how the experiment was conducted. In particular, there is no information on the amount of selenium given to each plant during the entire experiment. This information must be added. Additionally, the manuscript would benefit from moderate editing to improve English form.
Author Response
Dear Reviwer,
Thank you for your observation.
The different Se treatments were applied with a Steiner’s universal nutrient solution at the field water capacity, every 3 days.
As you suggested, it´s now indicated in the new version of the manuscript.
Best regards!